# Quantitative multi-pathway assessment of exposure to *Escherichia coli* for infants in Rural Ethiopia

**Yuke Wang**[1]*, **Yang Yang**[2], **Crystal M. Slanzi**[3], **Xiaolong Li**[4,5], **Amanda Ojeda**[6], **Fevi Paro**[4], **Loïc Deblais**[7,8], **Habib Yakubu**[1], **Bahar Mummed Hassen**[9], **Halengo Game**[10], **Kedir Teji Roba**[10,11], **Elizabeth Schieber**[12], **Abdulmuen Mohammed Ibrahim**[10], **Jeylan Wolyie**[13], **Jemal Yusuf Hassen**[10], **Gireesh Rajashekara**[7,8,14], **Sarah L. McKune**[4], **Arie H. Havelaar**[4], **Christine L. Moe**[1], **Song Liang**[15]

1 Hubert Department of Global Health, Rollins School of Public Health, Emory University, Atlanta, Georgia, United States of America, 2 Department of Statistics, Franklin College of Arts and Sciences, University of Georgia, Athens, Georgia, United States of America, 3 Department of Psychology, California State University, Los Angeles, Los Angeles, California, United States of America, 4 Emerging Pathogens Institute, University of Florida, Gainesville, Florida, United States of America, 5 Department of Environmental and Global Health, College of Public Health and Health Professions, University of Florida, Gainesville, Florida, United States of America, 6 Department of Microbiology and Cell Science, University of Florida, Gainesville, Florida, United States of America, 7 Department of Animal Sciences, CFAES Wooster, The Ohio State University, Wooster, Ohio, United States of America, 8 Global One Health initiative, The Ohio State University, Addis Ababa, Ethiopia, 9 College of Veterinary Medicine, Haramaya University, Dire Dawa, Ethiopia, 10 College of Health and Medical Sciences, Haramaya University, Dire Dawa, Ethiopia, 11 Department of Biobehavioral Health, Pennsylvania State University, University Park, Pennsylvania, United States of America, 12 dfusion Inc., Scotts Valley, California, United States of America, 13 College of Social Sciences and Humanities, Haramaya University, Dire Dawa, Ethiopia, 14 Department of Pathobiology, College of Veterinary Medicine, University of Illinois at Urbana-Champaign, Urbana, Illinois, United States of America, 15 Department of Environmental Health Sciences, School of Public Health and Health Sciences, University of Massachusetts, Amherst, Massachusetts, United States of America

* yuke.wang@emory.edu

**Data availability statement:** The datasets, including the high-resolution behavioral

## Abstract

In low- and middle-income countries (LMICs), enteric infections pose a significant threat to children's health. However, understanding the specifics of when, where, and how young children in LMICs are exposed to enteric pathogens and the roles of animal reservoirs, environmental media, and human hosts play during exposure remains limited. This study systematically evaluates infants' exposure to E. coli through various pathways in the rural Haramaya woreda of Ethiopia. Between June 2021 and June 2022, we conducted over 1300 hours of structured behavioral observations on 79 infants when aged 4–8 months (Timepoint 1) and 11–15 months (Timepoint 2). Enumerators recorded the infant's behavior related to exposure, including when and where it occurred, using a data collection system for behavioral data (Countee). Concurrently, we collected 1338 environmental samples from key contact interfaces between infants, other people, and the environment to test for E. coli. We used a competing hazard model for duration-based behaviors, an inhomogeneous Poisson point process model for frequency-based behaviors, and a left-censored lognormal distribution model for E. coli contamination levels.

observations and microbiological testing results, for this study can be found at GitHub (https://github.com/YWAN446/EXCAM).

**Funding:** This work was supported by the Bill Melinda Gates Foundation (OPP#007705 to SL and AH). The funders had no role in study design, data collection and analysis, decision to publish, or preparation of the manuscript.

**Competing interests:** The authors have declared that no competing interests exist.

The behavioral and environmental information was then integrated into an agent-based exposure model framework to quantify the exposure to E. coli through different pathways. The infant behavior, which altered the relative importance of different exposure pathways, changed as children grew older. Notably, we observed increased rates of touching behavior (e.g., touching fomites) and soil-pica, increased consumption of solid food, and more time spent on the bare ground at Timepoint 2. The major sources of exposure to E. coli were food and breastfeeding at Timepoint 1 and food and soil at Timepoint 2. This study provides insights for interventions to minimize infants' risk of exposure to fecal indicator bacteria, E. coli, and subsequent risk of enteric infections, including improved food handling practices, enhanced personal hygiene for breastfeeding caregivers, and education on the risk of soil-pica.

## Author summary

Together, unsafe water, unimproved sanitation, poor hygiene, and dirty environment make it easier for people, especially babies, in low and middle-income countries (LMICs) to catch enteric diseases. Generally, babies can get sick when they drink dirty water, eat contaminated food, do not wash their hands, play in dirty environment, or stay with other sick people. But what is the riskiest behavior for babies as they are growing up in LMICs is unknown. In this study, the authors examined the changing behavior of babies in rural Ethiopia and how those changes increase or decrease their chances of getting sick. The authors found that eating contaminated food and poor hygiene for breastfeeding mothers are the major concerns for little babies between 4 to 8 months old. As babies grow to 11 to 15 months old, additional risk is posed by their increasing abilities to crawl or walk around, which enables them to eat dirty soil in the environment. This research provides insights for how to reduce risk of disease for babies in the rural settings of Ethiopia.

## Introduction

Enteric diseases caused by enteric pathogens pose a substantial disease burden among people living in low- and middle-income countries (LMICs), with young children being particularly vulnerable [1]. Diarrheal diseases remain a leading cause of mortality among children under 5, accounting for approximately 443,832 deaths globally each year [2]. The highest mortality rates are observed in sub-Saharan Africa and South Asia [3]. Beyond acute diarrhea, enteric infections can lead to various short- and long-term health issues, such as malnutrition [4], physical [5] and cognitive [6] developmental challenges, and environmental enteric dysfunction [7], among others. Despite the well-recognized risk of these enteric pathogens, how, when, and where children in LMICs are exposed to these enteric pathogens [8] and quantitative descriptions of exposure to enteric microbes/pathogens through various pathways remain under-investigated and less understood. Multiple studies highlight the need to go beyond traditional water, sanitation, and hygiene (WASH) frameworks to include direct behavioral data. For example, time-activity studies in LMIC settings have shown that children are exposed through diverse behaviors such as geophagy [9], contact with animal feces, and frequent hand-to-mouth actions that may contribute more substantially to pathogen ingestion than contaminated drinking water alone [10,11].

There has been growing attention to young children's exposure to fecal contamination, often studied by fecal indicator, and enteric pathogens in LMICs due to the urgent need to mitigate associated adverse health effects [8,12,13]. Two large WASH intervention trials in rural Kenya and Zimbabwe found no improvements in child health outcomes, leading investigators to conclude that basic intervention–such as water chlorination and promotion, latrine improments and promotion, and handwashing stations with soap and hygiene promotion– were insufficient to reduce diarrhea and improve growth [14]. However, these studies offered limited insight into why the interventions failed. Only the Kenyan trial measured environmental contamination, and only through a few pathways (*E. coli* on children's hands, stored drinking water, and sentinel objects) [15]. Neither study assessed child behavior. Without examining how interventions affect exposure pathways and behavior, it is difficult to establish a mechanistic explanation for the lack of impact. Measuring fecal contamination along exposure pathways may help clarify how and when WASH interventions reduce disease risk. The SaniPath study has made a significant contribution to our understanding of exposure assessment in quantitatively assessing exposure to fecal indicator by combining information collected from structured behavioral observations and surveys with results from environmental and human microbiological sampling, primarily in urban settings involving both children and adults [16–19]. Other recent studies examined environmental exposure to enteric pathogens among infants and children under five in Kenya [20,21], Zimbabwe [22] and Bangladesh [23,24]. These studies have advanced our understanding of the complex pathways through which individuals are exposed to fecal contamination and enteric pathogens and provided insights that inform WASH intervention strategies. In addition, many previous researches [25–29] studied factors and pathways and improved assumptions for children's exposure to environmental contaminants and chemicals, which provided solid foundation of our modeling framework.

However, there are critical knowledge gaps about infants' exposure to enteric pathogens in rural LMIC settings. Questions about what the primary pathways of exposure are; how infants' behavior, including interactions with other people and the surrounding environment, mediate these pathways; how behavior changes with age; and how the fecal contamination and enteric pathogens are distributed in the rural environment of LMICs all remain largely unanswered. Given the vulnerability of infants to enteric diseases, closing these knowledge gaps is crucial.

Our study aims to systematically evaluate infants' exposure to *Escherichia coli* in the rural Haramaya woreda, Ethiopia. We employ a holistic modeling approach that integrates infants' behavior, interactions with other people and with their environment, and environmental contamination levels. The objectives of this paper are: (1) to comprehensively characterize and quantify infants' behavior, including interactions with other people and their surrounding environment, and (2) to quantify exposure to *Escherichia coli* from different sources through various pathways for two age groups of infants in rural Ethiopia.

## Materials and methods

### Ethics statement

This study was approved by the University of Florida Institutional Review Board (IRB201802987), the Institutional Health Research Ethics Review Committee (IHREC), Haramaya University, Ethiopia (IHRERC/091/2020), and National Research Ethics Review Committee, Ministry of Science and Higher Education (MoSHE), Ethiopia (SBA\117\7103\20). Written informed consent in local language (Afan Oromo) was obtained from the parent/guardian/head of household for each study infant.

## Study setting and population

The EXCAM study was nested within the broader *Campylobacter* Genomics and Environmental Enteric Dysfunction (CAGED) study. The CAGED longitudinal study assessed the prevalence, species composition, and genomic diversity of thermotolerant and non-thermotolerant *Campylobacter* spp. in infants, adults, livestock, and other reservoirs in the Haramaya woreda and used Whole Genome Sequencing data of *Campylobacter jejuni* isolates to attribute reservoirs of infection with these bacteria in children [30–34]. Building on a Health and Demographic Surveillance Site (HDSS) that covers 12 kebeles (i.e., the smallest administrative division in Ethiopia, similar to neighborhoods in urban areas) in the Haramaya woreda, established by Haramaya University [33], the CAGED study enrolled 115 newborn infants, that were randomly selected from 10 kebeles based on defined inclusion and exclusion criteria [33]. Demographics and socio-economic characteristics of the population in the study area were reported by Havelaar et al. [33] From this cohort, 79 newborns were randomly selected for the EXCAM study, which had two rounds of cross-sectional data collection: the first round occurred when infants were 4–8 months old (Timepoint 1), and the second round occurred when they were 11–15 months old (Timepoint 2). Among the 79 infants enrolled, three dropped out of the study (one infant deceased, and the other two moved out of the study area) before Timepoint 2.

## Data collection and sample testing

The design of our structured behavioral observations drew inspiration from the SaniPath study [16,17]. We customized the target behaviors and operational definitions for the context of the local rural setting, integrating insights from the CAGED study [31–33] and initial pilot field surveys. We adapted an application for mobile phones and tablets, "Countee", to capture high-resolution behavioral data on environmental contacts and interactions with other people [35,36]. Observations were conducted at two critical stages of early childhood development, initially in early infancy and subsequently in late infancy, to reflect evolving development, mobility, and interactions with surrounding environments. Prior to conducting any observations, four local enumerators were trained to record behavioral data during observations from both videos and live observations. We used an interobserver agreement and had the enumerators trained to 80% agreement in the natural environment (in infants' home) with a trained observer. Training continued until enumerators demonstrated competency and were able to record data reliably across observations. We conducted three scheduled training sessions and provided additional training whenever needed [37].

We collected over 1,300 hours of structured behavioral observation across all infants enrolled. At each timepoint, local enumerators conducted up to 10 hours of structured behavioral observations per infant (5 hours in the morning and 5 hours in the afternoon of the same day). This process yielded multi-dimensional time series data, recording the start and end time of duration-based behaviors (awake, bathing, drinking, and sleeping) and the timestamp of frequency-based behaviors (eating, mouthing, pica, and touching). Pica was defined as an event when any of the non-edible items is placed into the mouth such that it could be swallowed: feces, soil, or other. To ensure the duration-based behaviors are mutually exclusive, the "awake" behavior in this study was considered as awake but not bathing or drinking. Drinking, mostly breastfeeding, was recorded as a duration-based behavior as it is the most common ingestive behavior (with most time spent) for infants while rare instances of eating solid food was recorded as a frequency-based behavior. Data on where the behavior occurred, including location (within homestead and out of homestead) and compartments (carried by mother, carried by other, down on a surface with barriers, and down on the bare ground) was

also collected as duration-based records with start and end time. The "barriers" refers to the covering (e.g., a mat or blanket) put between the infant and surface (e.g., ground, bed) when the infant is laying down without the assistance of a person. The categories of behavior, compartment, and location defined in this study are sufficiently generalizable among the same age group.

After each structured observation, the sampling team, on the following day, collected samples from ten pre-identified human and environmental sample types (areola swabs of mothers, breast milk, mother handrinse, sibling (child) handrinse, infant handrinse, bathing water, drinking water, fomites, food, and soil) linked to the interfaces between infants and environment. Sampling locations in each household were selected based on information provided by the enumerators who conducted the behavioral observations. Soil samples were only collected at Timepoint 2 as we observed a limited mobility and contact with soil of infants at Timepoint 1. A total of 1338 human and environmental samples were collected, and all samples were transported in cold boxes containing ice packs to Haramaya University's laboratory within 6 hours of collection. Subsequently, samples were processed and tested for *E. coli* using two lab methods, depending on the nature of the sample [38]. EC MUG, a fluorometric approach, was used for drinking water, bathing water, handrinse, and the liquid extarcted from the fomites and areola swabs. Briefly, 180 ul of each handrinse, drinking water, and bathing water sample was added to 88 reaction wells per sample in a 96-deep well plate. Similarly, 180 ul of more concentrated samples (handrinses and fomite rinses) were added to 5 reaction wells per sample. 1.62 ml of EC-MUG broth was added to each reaction well, and the inoculated plate was sealed and incubated at 37°C for 48 hours. Each assay included four negative control wells with sterile peptone water and four positive control wells with an overnight *E. coli* culture. After incubation, 200 ul was withdrawn from each reaction well and transferred to a transparent 96-well plate, exposed to UV light, and the number of wells with fluorescence (*E. coli* positive) were counted. Tenfold serial dilutions in EC-MUG broth were tested for samples with *E. coli* concentrations too high to be quantified by the MPN approach described above.

Chromocult Coliform agar (Sigma Aldrich Millipore; St. Louis, MO) was used to measure *E. coli* in breast milk, resuspended solid food that was opaque, and soil samples. Samples were diluted tenfold in sterile 1X peptone water (pH 7), and 200 ul of undiluted and diluted samples were plated on the Cromocult agar. After 48 hour incubation at 37°C, purple colonies (*E. coli*) were counted. Each sample's *E. coli* concentration level was calculated based on the lab method (S1 Table). The details of laboratory methods and results can be found in Deblais et al. [38] The number of *E. coli* was quantified as Most Probable Number (MPN) for the EC MUG method and Colony Forming Units (CFU) for the Chromocult method. In this study, we assumed 1 MPN is equivalent to 1 CFU, and CFU was used as the unit for exposure assessment.

## Quantifying infants' behavior and fecal indicator

The duration-based behaviors and frequency-based behaviors were modeled in two steps. Similar to in Teunis et al., a combination of activity, compartment, and location was defined as a *state* for duration-based behaviors [16]. Among 32 possible *states*, only 21 *states* were observed in this study. Those *states* not observed were then omitted from the model. A change of *state* of a child was defined as a *transition*. Then, the sequence of duration-based behaviors can be considered a series of *transitions* between *states*. Such data can be visualized as a directed, weighted network with *states* as nodes and *transitions* as edges. The weight of an

edge represents the relative frequency of the corresponding *transition* observed. The competing hazard model in Teunis et al. was adapted to model the sequences of duration-based behaviors by timepoint [16]. From the start of the current *state*, all possible *transitions* with their own hazard functions compete for moving to the subsequent *state*. The rates of *transitions* were estimated using JAGS [39] and then used to generate Monte Carlo samples of sequences of duration-based behaviors in simulation studies.

When examining the occurrence of frequency-based behaviors (e.g., touching mother's hand) on a continuous timeline, we noted that the rate of the occurrences depends on the current *state* of the child (e.g., touching mother's hand is more likely when awake and carried by mother vs. when sleeping on a surface). Therefore, the frequency-based behavior was modeled as an inhomogeneous Poisson point process with the rate, $\lambda$, conditional on the current *state*. The expected occurrence rate of frequency-based behavior (*b*) during a certain *state* (*s*), $\lambda_{b,s}$, can be estimated as

$$\hat{\lambda}_{b,s} = \frac{N_{b,s}}{T_s},$$

where $N_{b,s}$ is the total number of occurrences for frequency-based behavior *b* during the time period that child is in the *state s* and $T_s$ is the cumulative length of the time period that child is in the *state s*. The inhomogeneous Poisson process of frequency-based behavior can be simulated by scaling (expanding or contracting) the time during different *states* with the estimated rates of the frequency-based behavior during those *states*. For example, given a frequency-based behavior has a rate of $\lambda = 3$ during a *state*, the length of time period of this *state* is expanded to three times its original length in the scaled timeline. Then, the *events* (i.e., occurrences of frequency-based behavior) can be simulated using a homogeneous Poisson process with $\lambda = 1$ at this scaled timeline and projected back to the original timeline (Fig 1). In addition, we conducted Wilcoxon signed-rank tests to compare the percent of time spent for sepecific duration-based activity, compartment, and location, as well as the rate of specific frequency-based activity between paired observations at Timepoint 1 and Timepoint 2.

For environmental samples, we conducted two-sample t-tests and Chi-squared tests to compare *E. coli* results (the concentration and proportion of positive samples) between two timepoints and examined the correlations between the log10 *E. coli* concentration levels of different sample types within the same household. The log10 *E. coli* concentration levels were modeled as a normal distribution with left-censoring, and samples with concentration levels

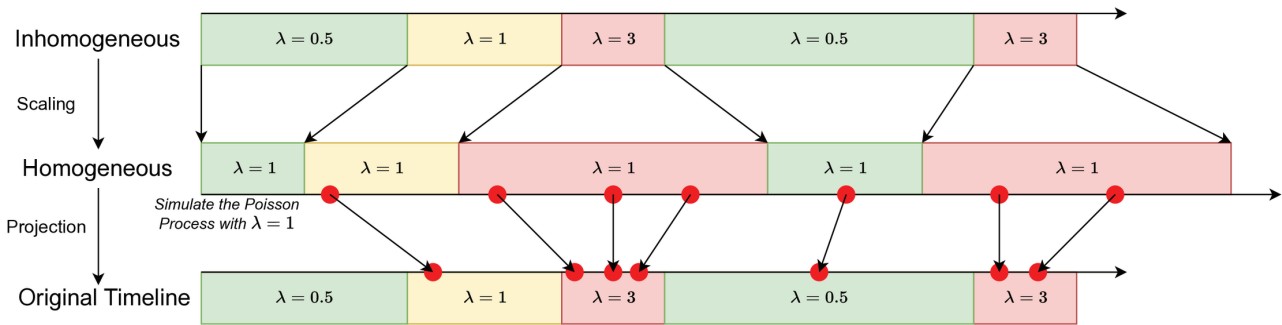

**Fig 1. Illustration of inhomogeneous Poisson process simulation.**

below the lower limit of detection (LLOD) were considered negative and thus left-censored. Parameters were estimated using the maximum likelihood method for each sample type at each timepoint.

## Multi-pathway exposure assessment

We adapted the Multi-pathway Agent-based Exposure Assessment model framework from the SaniPath study to track microbes from environmental and social *sources* through various pathways to ingestion by children [18,40]. In the current study, we consider direct pathway (i.e., *source* to *mouth*) and indirect pathway (i.e., *source* to *vehicle* to *month*). The environmental *sources* include fomites, soil, areola surface, breast milk, food, and drinking water, and the social *sources* include hands of the mother, other adults, and other children (e.g., siblings). The *sinks* were defined as destinations of microbes, including mouth and bathing water. The model includes two *vehicles*, child's hands and food, that can carry microbes from *sources* to *sinks*. We created seven modules to track *source*-specific microbe numbers transferred between *sources*, *vehicles*, and *sinks*, within a network structure:

1. **Hand Touching**: attachment and detachment of microbes on hands by touching hands of the mother, other adults, other children, and fomites.
2. **Mouthing**: microbe ingestion from mouthing hands of the mother, other adults, other children (siblings), infant itself, and fomites
3. **Bathing**: bathing water ingestion and microbe detachment from hands.
4. **Pica**: ingestion of soil and hand touching soil.
5. **Eating**: ingestion from eating solid food, possibly accompanied by hand touching of food.
6. **Breastfeeding**: breast milk ingestion and mouthing areola.
7. **Drinking Water**: ingestion of drinking water.

Simulated behavior sequences, including both duration-based behaviors and frequency-based behaviors, and estimated *E. coli* levels of various environmental compartments were input into the Exposure Model for each timepoint. Depending on the behavior (duration-based or frequency-based), the corresponding module calculates the number of microbes on hands and the microbes ingested, both as a vector of *E. coli* numbers from different original *sources* (Fig 2). The details of the exposure model, parameters, and assumptions were included in S1 Appendix. In this study, we generated microbe transfers during 10,000 typical child days (14 hours daytime assumed) for each timepoint. All simulations were run using R version 4.3.2. [41]. All codes for performing the data analyses and modeling are available at https://github.com/YWAN446/EXCAM.

## Results

### Quantifying infants' behavior

In this study, the infants' behavior, in terms of their environmental contact (e.g., touching fomites) and interactions with other people (e.g., touching mother's hands), evolved with age. Table 1 shows the differences in infants' behavior between two timepoints (Timepoint 1: 4–8 months old vs. Timepoint 2: 11–15 months old). The infants at Timepoint 1 spent more time sleeping (p<0.001) during the day compared to those at Timepoint 2. Bathing behavior was infrequent and short for both timepoints. Only 30.4% of study infants at Timepoint 1 and 57.9% at Timepoint 2 bathed during our observations. And the average length of bathing is

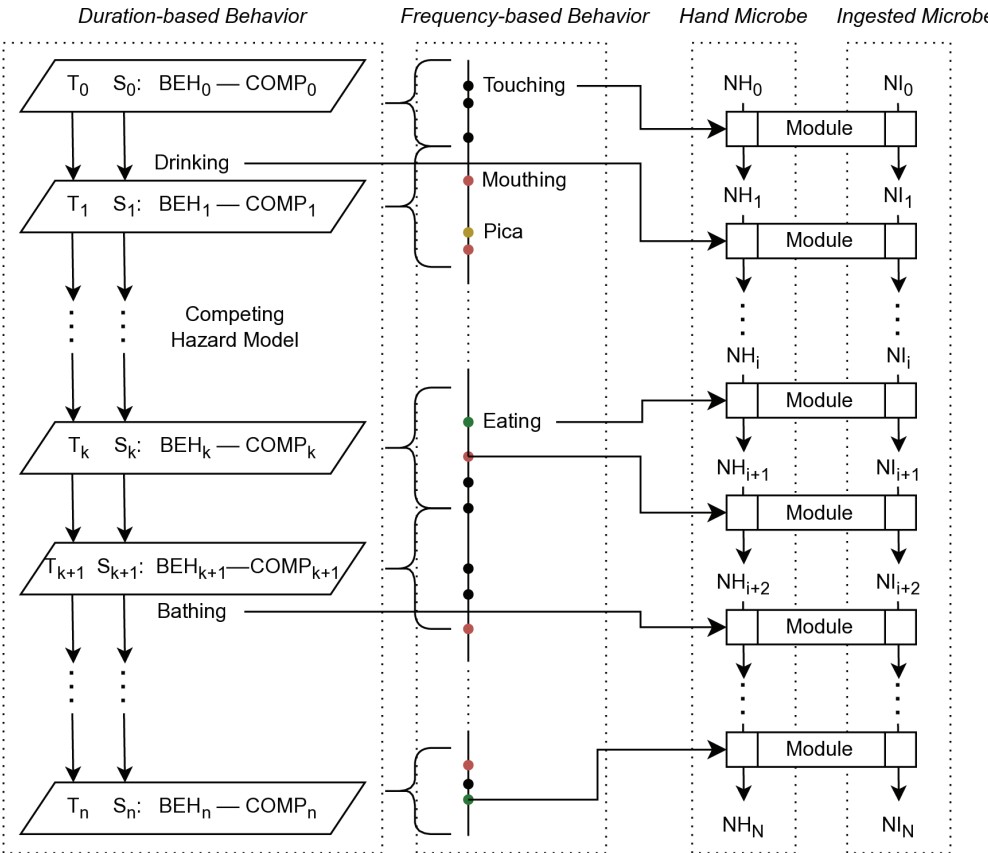

**Fig 2. Exposure Model structure.** Time is denoted as $T$. State ($S$) is a combination of behavior ($BEH$) and compartment ($COMP$). Competing Hazard model will generate the next time ($T$) and state ($S$) until $T_n - T_0 > 14$ hours. Modules determined by behavior sequences calculate the microbes on hands ($NH$) and microbes ingested ($NI$).

5.8 mins at Timepoint 1 and 2.1 mins at Timpoint 2. Rates (the number of times per hour) of eating, pica, and touching increased substantially (p<0.001) while the rate of mouthing decreased from Timepoint 1 to Timepoint 2.

The nature of infant contact with their environment also changed between two timepoints. Infants at Timepoint 1 rarely spent any time on the bare ground, in contrast to Timepoint 2, when infants spent 16.4% of time on the bare ground (e.g., crawling). During 54.7% of the observation period at Timepoint 1, infants were carried either by mothers or other caregivers; this decreased to 33.9% of the observation period at Timepoint 2. Infants in the study rarely went outside the homestead, defined as a cluster of buildings accommodating an extended family and their small livestock (chickens, small ruminants).

S1 and S2 Figs present the variations in the proportions of time spent engaging in duration-based behaviors and the rates of frequency-based behaviors between individual infants. Table 2 shows the rates of frequency-based behaviors in each subcategory by timepoint. The rate of consuming solid food and drinking water increased while the rate of breastfeeding slightly decreased from Timepoint 1 to Timepoint 2. At Timepoint 2, infants touched environmental fomites more often and showed a greater tendency towards soil-pica than at Timepoint 1. In addition, we examined the patterns in the sequences of behavior. S3–S6 Figs

**Table 1. Descriptive statistics of child behavior during structured observations at two timepoints.**

| | Timepoint 1 (May–Dec 2021) | Timepoint 2 (Dec 2021–June 2022) | Wilcoxon Signed-rank Test p-value |
|---|---|---|---|
| # subjects | 79 | 76 | |
| mean age (range) in days | 184 (132–263) | 371 (338–466) | |
| duration of observation (hours) | 702 | 608 | |
| # duration-based activity transitions | 3315 | 2787 | |
| average % of time spent awake | 60.9 | 76.0 | <0.001 |
| average % of time spent bathing | 0.3 | 0.3 | 0.106 |
| average % of time spent drinking | 16.9 | 11.7 | <0.001 |
| average % of time spent sleeping | 21.9 | 12.1 | <0.001 |
| # frequency-based activities | 23,677 | 30,801 | |
| rate of eating (times per hour) | 0.35 | 2.20 | <0.001 |
| rate of mouthing (times per hour) | 13.13 | 10.69 | <0.001 |
| rate of pica (times per hour) | 0.05 | 0.67 | <0.001 |
| rate of touching (times per hour) | 20.07 | 37.11 | <0.001 |
| # compartment transitions | 2688 | 3632 | |
| average % of time spent carried by mother | 40.3 | 27.0 | <0.001 |
| average % of time spent carried by other | 14.4 | 6.9 | <0.001 |
| average % of time spent on the bare ground | 0.2 | 16.4 | 0.005 |
| average % of time spent on a surface with barriers | 45.1 | 49.8 | <0.001 |
| # location transitions | 47 | 149 | |
| average % of time spent out of homestead | 0.4 | 2.2 | <0.001 |
| average % of time spent within homestead | 99.6 | 97.8 | <0.001 |

**Table 2. Rate of subcategory duration-based behaviors and drinking behavior during the structured observations at two timepoints.**

| Rate of *events* (times per hour) | Timepoint 1 (4–8 months old) | Timepoint 2 (11–15 months old) | Wilcoxon Signed-rank Test p-value |
|---|---|---|---|
| drinking breast milk | 1.67 | 1.45 | <0.001 |
| drinking water | 0.13 | 0.23 | 0.715 |
| drinking other | 0.22 | 0.28 | 0.003 |
| eating injera | 0.17 | 0.98 | <0.001 |
| eating raw produce | 0.001 | 0.04 | 0.005 |
| eating other | 0.18 | 1.18 | <0.001 |
| mouthing own hands | 8.84 | 4.55 | <0.001 |
| mouthing mother's hands | 0.59 | 0.22 | <0.001 |
| mouthing other adult's hands | 0.28 | 0.08 | <0.001 |
| mouthing other child's hands | 0.21 | 0.13 | 0.053 |
| mouthing fomites | 3.21 | 5.72 | <0.001 |
| pica feces | 0 | 0.002 | NA |
| pica soil | 0.003 | 0.36 | <0.001 |
| pica other | 0.04 | 0.31 | <0.001 |
| touching mother's hands | 8.80 | 9.33 | 0.287 |
| touching other adult's hands | 3.99 | 4.41 | 0.258 |
| touching other child's hands | 3.38 | 5.17 | <0.001 |
| touching livestock | 0.02 | 0.09 | <0.001 |
| touching fomites | 3.88 | 18.10 | <0.001 |

show the activities and compartments for sequences of duration-based behaviors at two time-points. It is visually obvious that sleeping occurred uniformly during the day at Timepoint 1, while at Timepoint 2 infants tended to sleep around noon.

The two-dimensional behavior sequences were visualized as directed weighted networks with *states* (combinations of activity and compartment) as nodes and *transitions* between *states* as edges (S7 Fig). The rates of *transitions* were estimated in the competing hazard model. S8 and S9 Figs show the marginal and conditional estimated rates of frequency-based

behaviors at Timepoint 1 and Timepoint 2, respectively. Mouthing baby's hands and touching caregiver's hands occurred across different *states*. Touching and mouthing fomites commonly occurred when the infants were awake and down on a surface with barriers or down on the bare ground.

### Quantifying fecal indicator

The percent of *E. coli* positive samples and the mean *E. coli* concentration of positive samples, measured on a log10 scale, by sample type and timepoint are presented in Table 3, adapted from Deblais et al. [38] The percent of *E. coli* positive between the two timepoints was significantly different for breast milk (p=0.003), infant handrinse (p=0.001), and drinking water (p=0.002). The mean log10 *E. coli* concentrations were significantly different between the two timepoints for infant handrinse (p=0.003) and fomites (p=0.016). Infant handrinse samples had both more *E. coli* detection and higher *E. coli* concentrations at Timepoint 2 compared to at Timepoint 1. After assigning the LLOD to the *E. coli* concentration for the negative samples, we calculated the correlations between *E. coli* contamination levels (in log10 scale) of different sample types within the same study household (S10 Fig). We found that the *E. coli* contamination level of infant handrinse was correlated with the *E. coli* contamination levels of mother handrinse (Spearman's $\rho$=0.39), sibling handrinse (Spearman's $\rho$=0.35), areola swabs (Spearman's $\rho$=0.28), and fomites (Spearman's $\rho$=0.29). The *E. coli* contamination level of mother handrinse was correlated with the *E. coli* contamination levels of infant handrinse (Spearman's $\rho$=0.39), sibling handrinse (Spearman's $\rho$=0.43), drinking water (Spearman's $\rho$=0.22), areola swabs (Spearman's $\rho$=0.27), and fomites (Spearman's $\rho$=0.23). More detailed correlation analysis results can be found in Deblais et al. [38] S2 Table shows the estimated parameters of normal distribution for *E. coli* concentration (in log10 scale) which were used in the exposure assessment. As we did not collect soil samples at Timepoint 1, the estimated parameters from soil samples at Timepoint 2 were used for the exposure assessment at both Timepoint 1 and Timepoint 2.

### Exposure to fecal indicator

Fig 3 shows the average relative contributions of different social and environmental *sources* to the *ingestion* of fecal indicator (measured as *E. coli*) across 10,000 simulations at the two timepoints (Timepoint 1: 4–8 months old vs. Timepoint 2: 11–15 months old), and S11 Fig shows variation in exposure by *sources* and timepoint. Food and breastfeeding were the major contributors, "dominant pathways" [18], to the exposure to *E. coli* for infants at Timepoint 1, and as the infants grew older, food and soil were the major contributors at Timepoint 2. The average exposure per day, indicated by the color in Fig 3, was higher at Timepoint 2 compared to at Timepoint 1. Fig 4 shows the "fecal microbe transfer networks" [18] for infants. Direct pathways such as eating, breastfeeding, soil-pica, and mouthing fomites resulted in higher exposure to *E. coli*. In contrast, hand contact with environmental compartments (i.e., touching fomites or soil) and interactions with other people (i.e., touching the hands of the mother, other adults, and other children) did not substantially increase the *E. coli* on infants' hands (even decreased it at Timepoint 2). The indirect environment-hand-mouth pathways were responsible for a relatively small proportion of the total exposure to *E. coli*.

### Discussion

This study represents a significant component of our broader efforts to systematically assess the exposure to fecal indicator bacteria, and ultimately to important enteric pathogens for

**Table 3. Detection and concentration of *E. coli* in environmental samples at two timepoints.**

| Sample Type | Timepoint | N | Percent of *E. coli* positive | Log10 Mean (Range) Concentration in Positive Samples | Unit |
|---|---|---|---|---|---|
| Areola Swab | 1 | 78 | 16.7 | 1.11 (0.68–1.53) | per swab |
|  | 2 | 75 | 18.7 | 2.21 (0.68–3.32) | per swab |
| Breast Milk* | 1 | 51 | 21.6 | 1.95 (0.70–2.86) | per mL |
|  | 2 | 69 | 2.9 | 1.35 (0.70–1.60) | per mL |
| Mother Handrinse | 1 | 79 | 81.0 | 3.64 (1.34–5.05) | per pair of hands |
|  | 2 | 76 | 84.2 | 4.26 (1.34–6.05) | per pair of hands |
| Sibling Handrinse | 1 | 64 | 89.1 | 5.29 (1.34–7.05) | per pair of hands |
|  | 2 | 62 | 85.5 | 4.33 (1.34–6.05) | per pair of hands |
| Infant Handrinse*† | 1 | 79 | 51.9 | 2.63 (1.34–4.05) | per pair of hands |
|  | 2 | 76 | 77.6 | 3.63 (1.34–5.05) | per pair of hands |
| Bathing Water | 1 | 24 | 91.7 | 2.40 (-0.96–3.74) | per mL |
|  | 2 | 21 | 81.0 | 0.71 (-0.96–1.74) | per mL |
| Drinking Water* | 1 | 75 | 77.3 | 0.08 (-0.96–1.32) | per mL |
|  | 2 | 76 | 51.3 | -0.30 (-0.96–0.75) | per mL |
| Fomites† | 1 | 129 | 63.6 | 2.07 (0.63–3.27) | per sponge |
|  | 2 | 145 | 56.6 | 3.55 (0.63–5.27) | per sponge |
| Food | 1 | 34 | 35.3 | 2.51 (0.63–3.39) | per gram |
|  | 2 | 68 | 25.0 | 3.35 (0.63–4.27) | per gram |
| Soil | 1 | 0 |  |  |  |
|  | 2 | 57 | 98.2 | 5.94 (3.57–7.35) | per boot sock |
| Total | 1 | 613 | 58.7 |  |  |
|  | 2 | 725 | 55.6 |  |  |

*The difference in the percent of samples *E. coli* positive is significant between the two timepoints by Chi-sqaured tests with a significance level of 0.05. †The difference in mean log10 *E. coli* concentrations is significant between the two timepoints by two-sample t-tests with a significance level of 0.05. Food includes injera, cow milk, rice, macaroni, biscuits, etc. Fomites include tea cups, baby bottles, mobile phones, etc.

infants in rural regions of Ethiopia. Our primary aim is to gain insights into how children's behavior, including interactions with other people and their surrounding environment, modulates their exposure to *E. coli* as the fecal indicator bacteria. This study is notable for being the first to collect and analyze extensive, high-resolution, multi-dimensional behavior sequence data for exposure assessment in rural settings in LMICs. Along with environmental and human microbiological samples as input, an agent-based exposure model can perform a comprehensive assessment to investigate infants' exposure to fecal indicator and enteric pathogens through multiple pathways. Overall, though less frequently consumed, solid food is the most dominant pathway that contributes the highest exposure to *E. coli*, which is similar to our findings for children 5–12 years old and adults in many countries around the globe [19]. Several key findings of our study include:

1. Changes in infant behavior as they grow alter the relative contributions of various environmental exposure pathways.
2. Infants could ingest fecal indicator on mothers' hands, nipples, and areola areas through breastfeeding.
3. Contaminated soil is an important environmental *source* of fecal exposure through infants' soil-pica behavior.

The infants' behavior, including interactions with other people and the surrounding environment, evolves rapidly as they grow [42,43]. For the first few months after birth, infants'

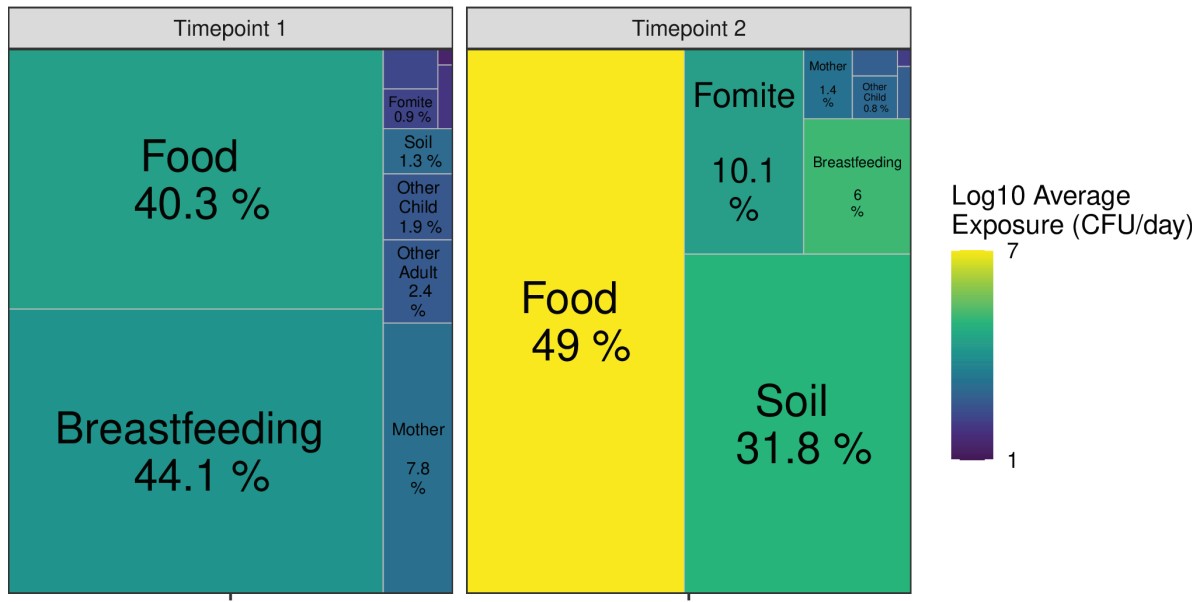

**Fig 3. Average contribution (in percentage) of exposure to *E. coli* over 10,000 simulated child days by source and timepoint.** The color represents the average daily exposure from a specific *source*.

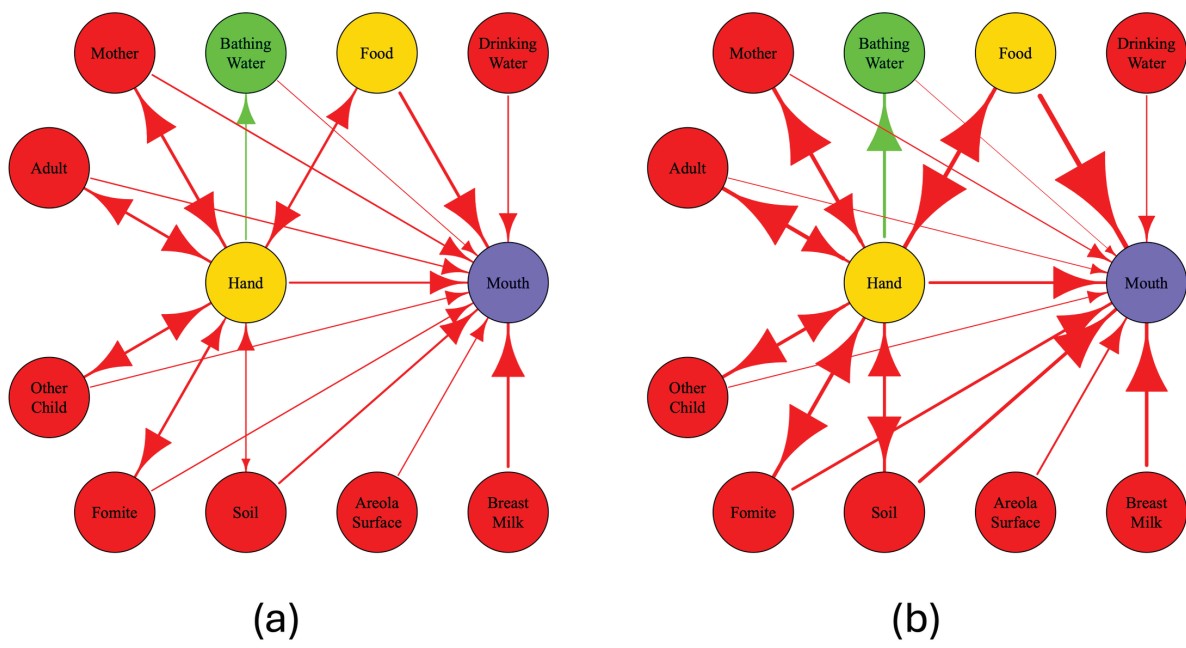

**Fig 4. Fecal microbe transfer networks averaged over 10,000 simulated child days for children at Timepoint 1, subfigure (a), and Timepoint 2, subfigure (b).** The size of arrows and edges are proportional to the log10 of the average numbers of fecal microbes transferred (for 10,000 simulated child days). The color of nodes represents their role in the network. Red: *sources*; yellow: *vehicles* (can be source and sink); green: *sinks* (remove contamination); purple: *ingestion*. Food is considered as a *vehicle* and *source*, while bathing water is considered as a *sink* and *source*.

behavior is predominantly, if not purely, driven by their biological needs, such as eating, drinking, sleeping, defecating, and seeking comfort [44]. Breastfeeding (due to breast milk being the primary food source) and sleeping occur frequently during the day without a schedule [45]. Their movements around the environment are generally restricted (within the private domain, such as households) and passive (carried by caregivers), while their interactions with caregivers (and siblings) are frequent and close. During this stage of life, poor hygiene behavior of caregivers (e.g., lack of handwashing) and a lack of cleanliness within the household (e.g., dirty toys for mouthing, household surfaces, and food preparation surfaces), due to limited water supply and unimproved sanitation, could have a significant impact on the risk of exposure to fecal contamination and enteric pathogens. During late infancy, they develop basic mobility (i.e., crawling or walking), leading to more active and frequent interaction with the environment both in private (i.e., within homestead) and public domains (i.e., out of homestead) [43]. Studies [46,47] reported that children aged 6–23 months were found to be much more likely to experience diarrhea compared to those under 6 months, which could be driven by their changing behavior. In our study, the infants' hands were observed to be significantly dirtier (p=0.003) at Timepoint 2 (11–15 months old) compared to Timepoint 1 (4–8 months old). We found a lower frequency of touching fomites for infants at Timepoint 1 compared to Timepoint 2. This could be caused by the limited mobility of young infants and longer time carried by mother or other at Timepont 1. We also observed a 120-times increase in the average rate of soil-pica, a high-risk behavior, at Timepoint 2. In addition, the infants observed started consuming more solid food to supplement the breast milk and the wider range of food that they ingest increases the risk of exposure to contaminated food. These changes in infant behavior alter their exposure profile, which is the relative importance of different environmental pathways contributing to exposure to fecal contamination and enteric pathogens. Information on the dominant environmental exposure pathways is critical for targeting sanitation and hygiene interventions and evaluating their effectiveness. For example, the soil pathway was not a major risk for infants in rural Ethiopia at Timepoint 1, but it became a dominant exposure pathway for *E. coli*, along with the solid food pathway, at Timepoint 2. As a result, different interventions would be recommended to reduce the infants' exposure substantially enough to lead to measurable health impacts.

Breast milk is universally valued for its nutritional and immunologic benefits for infants. Surprisingly, at Timepoint 1, 21.6% of breast milk samples were contaminated with *E. coli* and breastfeeding was one of the most important *sources* of exposure to *E. coli* for infants. There is no evidence that *E. coli* is secreted in breast milk [48]. In 2016, Nakamura et al. reported an outbreak of extended-spectrum $\beta$-lactamase (ESBL) producing *E. coli* through breast milk sharing in a neonatal intensive care unit [49]. The breast milk was exclusively contaminated from the donor mother's left breast, which she reported was not sufficiently cleaned due to a sore nipple [49]. Therefore, it is likely that the *E. coli* contamination in the breast milk samples in our study originated from the mothers' nipples and areola areas, where we detected a similar percent of positive for *E. coli* (16.7% at Timepoint 1 and 18.7% at Timepoint 2). Discussions with members of the local field team also indicated that breastfeeding practices often overlook personal hygiene measures in rural communities in Ethiopia. For breastfeeding, latching or unlatching a young infant requires the mother to put her hands on her breast, as well as in and around the infant's mouth. Many mothers, possibly due to a lack of awareness or understanding of the importance of hygienic practices [50], feed their babies without first properly cleaning the breasts or their hands. Instances were noted during our structured observations, when a mother merely wiped the nipple with her clothes or hands before feeding the child. Our findings suggested that behavioral interventions targeting personal hygiene

practices, such as handwashing and cleaning the nipple and areola area, before breastfeeding should be promoted in rural Ethiopia and areas with similar situations.

Unlike in urban settings, where concrete floor surfaces are common in households, most floors in rural households are composed of soil and cemented mud (a mixture of soil and livestock feces). In our study, fecal contamination was widely distributed with elevated *E. coli* concentration levels in the soil samples collected in the household compound. The soil within and outside households could be contaminated with fecal material from both humans and animals. In rural communities, many households do not have latrines [50] and open defecation in backyards or fields is common. Meanwhile, humans and livestock often coexist under the same roof, particularly at night, resulting in animals defecating and urinating in the shared living space. Ercumen et al. reported higher *E. coli* levels were associated with the ownership of free-roaming animals [51]. In such a living environment, unintentionally ingesting soil through hand-mouthing or intentional ingestion through soil-pica could result in a large amount of exposure to fecal contamination and enteric pathogens for infants. Previous studies [22,52–54] and the behavioral observation data from this study showed that soil-pica is not a rare behavior among young children. Enhancing household living environments and implementing effective animal fecal management should be considered to protect young children in rural Ethiopia.

This study had several strengths compared to previous exposure assessment research. Structured behavioral observation data with much higher resolution were collected with 60,580 infant behavior events in 1310 hours compared to our previous SaniPath study in Accra, Ghana, which only had 1846 events in 500 hours [16]. The behavioral data in this study were collected at the very early stages of infants' lives, when they are most vulnerable to enteric diseases. Such rich behavioral data enable us to examine the behavior differences in detail between age groups and build a more complex behavior model with two layers for duration-based and frequency-based behaviors. Environmental samples of different types were collected within the same household right after the behavioral observation, and the enumerators conducting those observations informed the sampling locations of environmental samples. This type of matched design enabled us to identify the positive correlation between the fecal indicator levels on the hands of infants, mothers, and siblings, mothers' skin (e.g., areola surface), and fomite surfaces. In addition, this study included previously overlooked exposure pathways like breastfeeding, mouthing caregivers' hands, and soil-pica and revealed surprising findings about the important role of these dominant exposure pathways for infants in rural Ethiopia. The model developed here can be generalized to quantify exposure to a variety of enteric pathogens and even chemicals in the environment.

Yet, we acknowledge some limitations underlying the study. Capturing the full spectrum of infants' behavior and exposure in their environments presents significant challenges due to the dynamic and complex nature of daily activities and interactions. Recording only the infants' behavior in pre-defined behavior types that are frequently observed may not fully account for occasional but high-risk exposure pathways, such as touching animal fecal matter. Also, we were not able to observe the infants during specific time periods, such as right after waking up in the morning or before going to sleep at night. This could lead to potential underestimation or oversight in identifying the contributions from specific pathways to exposure to fecal contamination and enteric pathogens. Another limitation of our study comes from the logistical and operational challenges exacerbated by unforeseen external circumstances. The behavioral observations and environmental sample collection were originally planned during the developmental windows of infants ages 1–3 months old and 6–9 months old. However, the concurrent impact of the COVID-19 pandemic and geopolitical tension within the country delayed the delivery of necessary lab supplies and equipment to

Haramaya University and disrupted our fieldwork schedule. Despite the delay in data collection, the adjusted timeline windows (spanning ages 4–8 months and 11–15 months) still provided valuable insights into the behavior patterns of infants in rural Ethiopia early in their life and associated exposure to fecal contamination and enteric pathogens. Infants between 4 to 8 months old (at Timepoint 1) predominantly rely on breastfeeding and exhibit limited mobility, aligning closely with our initial proposed age groups. The observation at Timepoint 2 (11–15 months old) was more distinct from the observation of the younger infants and thus offered a unique opportunity to understand evolving behavioral patterns and exposure to fecal indicator associated with growing mobility and a more diverse diet. Also, our environmental sample collection (based on a rolling enrollment) cannot systematically examine the environmental contamination risk driven by seasonality which has been reported by previous studies [55,56]. In addition, a direct comparison of the behavior data between this study with our previous studies is difficult because the study design, including subjects, settings, behavior definitions, training, and data collection tool, has evolved over time.

The findings from this study have important policy implications. In rural Ethiopia, even young infants consume solid food, such as injera (i.e., a traditional Ethiopian flatbread made from teff flour), which commonly has fecal contamination. Contamination of injera is most likely introduced during food handling and storage rather than its cooking process which involves a high temperature. Behavioral interventions to promote handwashing before eating and feeding for both infants and their caregivers and better food handling and storage practices are recommended. This research also highlights the need for targeted sanitation and hygiene improvement initiatives and education for expecting and breastfeeding mothers. Such efforts should include the separation of human and animal spaces, proper disposal of human and animal feces, training on animal husbandry, maintaining household cleanliness, and promoting handwashing and personal hygiene before breastfeeding and infant food preparation. These improvements are essential for reducing potential fecal-oral transmission of enteric pathogens. In addition, information about the behavioral evolution of infants during development should be integrated into maternal and child health educational and intervention programs. As infants grow and become more mobile, culturally appropriate strategies to prevent soil ingestion need to be developed and implemented. While human behavior and environment are dynamic and location specific, which means we need to collect new data for each assessment, the overall multi-pathway exposure assessment design (e.g., categories of behavior, environmental pathways) can be very similar across settings and countries, making the data collection and modeling framework generalizable.

Moving forward, our results suggest several directions for future research. First, expanding these research methodologies to explore exposure to specific enteric pathogens, with different ecological and biological characteristics, will offer a more holistic view of exposure risks and their mitigation. Second, conducting an in-depth assessment of the challenges to promoting good personal hygiene for mothers before breastfeeding may be beneficial. Developing community-based educational and behavioral intervention programs that are culturally sensitive and socially acceptable to the community, is essential for adopting recommended practices. Finally, the exposure assessment model developed here can be used to conduct simulation studies evaluating the efficacy of specific behavioral interventions, such as campaigns to promote handwashing and education on hygienic breastfeeding practices, and could guide the design and implementation of locally sustainable and effective intervention strategies.

## Conclusion

This study highlights the crucial role of infant behavior, including interactions with other people and their immediate living environment, in understanding how infants are exposed to fecal contamination and enteric pathogens in rural settings of LMICs. We developed an integrated exposure assessment model framework that identified and quantified the changes in the exposure profile (e.g., dominant exposure pathways) between infants at two age periods. The findings of high exposure to *E. coli* from solid food, breast milk, and soil in this rural study setting are critical evidence for reducing infants' exposure risks by promoting safe food handling and storage, improved personal hygiene before breastfeeding, education on the risk of soil ingestion and measures to prevent soil-pica.

## Supporting information

**S1 Fig. Proportion of time spent for duration-based behaviors by Timepoint.** Boxplots show distributions of the proportions of time spent on specific activities, compartments, or locations among infants. The central line in the box indicates the median value, while the box limits indicate the first and third quartile. The points outside of the box indicate outliers. (TIF)

**S2 Fig. Rate (times per hour) of frequency-based behaviors by Timepoint.** Boxplots show distributions of the rates of frequency-based behaviors among infants. The central line in the box indicates the median value, while the box limits indicate the first and third quartile. The points outside of the box indicate outliers. (TIF)

**S3 Fig. Observed activity sequences at Timepoint 1.** The morning and afternoon sessions were combined for each infant. The x-axis shows the time and the y-axis shows the age of infants, which is sorted in ascending order (from bottom to top). "NA" represents the time period not observed. (TIF)

**S4 Fig. Observed activity sequences at Timepoint 2.** The morning and afternoon sessions were combined for each infant. The x-axis shows the time and the y-axis shows the age of infants, which is sorted in ascending order (from bottom to top). "NA" represents the time period not observed. (TIF)

**S5 Fig. Observed compartment sequences at Timepoint 1.** The morning and afternoon sessions were combined for each infant. The x-axis shows the time and the y-axis shows the age of infants, which is sorted in ascending order (from bottom to top). "NA" represents the time period not observed. (TIF)

**S6 Fig. Observed compartment sequences at Timepoint 2.** The morning and afternoon sessions were combined for each infant. The x-axis shows the time and the y-axis shows the age of infants, which is sorted in ascending order (from bottom to top). "NA" represents the time period not observed. (TIF)

**S7 Fig. Observed behavior transition networks.** (a) Observed behavior transition network at Timepoint 1 (6050 transitions); (b) observed behavior transition network at Timepoint 2 (6568 transitions). The text inside the node shows the activity. The color of the node represents the compartment: light gray is Carried by Mother; dark gray is Carried by Others;

pink is Down on a Surface with Barriers; red is Down on the Bare Ground. For the location, the nodes above the horizontal line are within homestead, while those below the horizontal line are out of homestead. Arrows indicate transitions between states (i.e., combinations of activity, compartment, and location), and strengths (numbers of times the transition was observed) indicated by arrow width and shade (darker arrows indicate higher frequency). (TIF)

**S8 Fig. Estimated rate (times per hour) of frequency-based behaviors conditional on the state at Timepoint 1.** The color represents the rate: darker colors show higher rates. (TIF)

**S9 Fig. Estimated rate (times per hour) of frequency-based behaviors conditional on the state at Timepoint 2.** The color represents the rate: darker colors show higher rates. (TIF)

**S10 Fig. Correlation matrix for log10 scale *E. coli* concentration levels between different sample types.** In the correlation matrix, plots in the bottom left half show the scatterplots of log10 scale *E. coli* concentration levels for pairs of sample types. Plots on the diagonal line show the histogram of log10 scale *E. coli* concentration levels by sample type. Plots on the top right half show the Spearman correlation coefficients of log10 scale *E. coli* concentration levels between different sample types. The stars in the figure define the level of significance. * = 0.05, ** = 0.01, *** = 0.001. (TIF)

**S11 Fig. Exposure to *E. coli* by source and timepoint.** The bar charts show the fraction of simulated days that children are exposed, and boxplots show the estimated daily dose of *E. coli* ($\log_{10}$ CFU/day). (TIF)

**S1 Appendix. The details of the exposure model, parameters, and assumptions.** (PDF)

**S1 Table. Lab methods for different sample types.** For the areola swab, fomites, handrinse, and soil samples, we reported elution volume as collection volume. The approximate surface area of bootsock samples is between 8000 and 10000 $cm^2$. (PDF)

**S2 Table. Estimated parameters of normal distributions for log 10 scale *E. coli* concentrations at two timepoints.** (PDF)

## Acknowledgments

We thank Ballo Mummed, Belisa Usmael Ahmedo, Efrah Yusuf, Ibsa Ahmed, Ibsa Fayo Abrahim, Kedir Hassen, Meri Usmail, Mussie Brhane, and Yenenesh Demisie Weldesenbet from Haramaya University for their substantial work conducting behavioral observations, sample collections, and lab analyses. We thank Dr. Peter Teunis at Emory University for his insights into developing behavior and exposure models. We thank Sarah Durry at Emory University for her insights into the study design in Ethiopia. We thank Shaolin Xiang at the University of Georgia for identifying the missing values in the behavioral data. At last, special thanks to my children, Angelina Wangpan and Abigail Wangpan, for teaching me the basics of infant behavior.

## Author contributions

**Conceptualization:** Yuke Wang, Yang Yang, Habib Yakubu, Gireesh Rajashekara, Sarah L. McKune, Arie H. Havelaar, Christine L. Moe, Song Liang.

**Data curation:** Xiaolong Li.

**Formal analysis:** Yuke Wang, Yang Yang, Xiaolong Li.

**Funding acquisition:** Song Liang.

**Investigation:** Crystal M. Slanzi, Amanda Ojeda, Loïc Deblais, Bahar Mummed Hassen, Halengo Game, Kedir Teji Roba, Elizabeth Schieber, Abdulmuen Mohammed Ibrahim, Jeylan Wolyie, Jemal Yusuf Hassen, Gireesh Rajashekara, Sarah L. McKune.

**Methodology:** Yuke Wang, Yang Yang, Crystal M. Slanzi, Loïc Deblais, Elizabeth Schieber, Gireesh Rajashekara, Sarah L. McKune.

**Project administration:** Amanda Ojeda.

**Software:** Yuke Wang.

**Supervision:** Arie H. Havelaar, Christine L. Moe, Song Liang.

**Writing – original draft:** Yuke Wang.

**Writing – review & editing:** Yang Yang, Crystal M. Slanzi, Xiaolong Li, Amanda Ojeda, Fevi Paro, Loïc Deblais, Kedir Teji Roba, Elizabeth Schieber, Gireesh Rajashekara, Sarah L. McKune, Arie H. Havelaar, Christine L. Moe, Song Liang.

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
