## [Decision Letter · Decision Letter 0]

PNTD-D-24-01785

Quantitative Multi-pathway Assessment of Exposure to Fecal Contamination for Infants in Rural Ethiopia

Dear Dr. Wang,

Thank you for submitting your manuscript to PLOS Neglected Tropical Diseases. After careful consideration, we feel that it has merit but does not fully meet PLOS Neglected Tropical Diseases's publication criteria as it currently stands. Therefore, we invite you to submit a revised version of the manuscript that addresses the points raised during the review process.

Please submit your revised manuscript within 60 days Apr 15 2025 11:59PM. If you will need more time than this to complete your revisions, please reply to this message or contact the journal office at plosntds@plos.org. Please include the following items when submitting your revised manuscript:

We look forward to receiving your revised manuscript.

Kind regards,

Josh M Colston, Ph.D.

Academic Editor

Ana LTO Nascimento

Section Editor

Shaden Kamhawi

co-Editor-in-Chief

Paul Brindley

co-Editor-in-Chief

Additional Editor Comments:

Please address the reviewers' comments point by point.

Journal Requirements:

At this stage, the following Authors/Authors require contributions: Yuke Wang, Yang Yang, Crystal M. Slanzi, Xiaolong Li, Amanda Ojeda, Fevi Paro, Loïc Deblais, Habib Yakubu, Bahar Mummed Hassen, Halengo Game, Kedir Teji Roba, Elizabeth Schieber, Abdulmuen Mohammed Ibrahim, Jeylan Wolyie, Jemal Yusuf Hassen, Gireesh Rajashekara, Sarah L. McKune, Arie H. Havelaar, Christine Moe, and Song Liang. Please ensure that the full contributions of each author are acknowledged in the "Add/Edit/Remove Authors" section of our submission form.

4) We do not publish any copyright or trademark symbols that usually accompany proprietary names, eg ©,  ®, or TM  (e.g. next to drug or reagent names). Therefore please remove all instances of trademark/copyright symbols throughout the text, including:

- © on page: 2.

5) Please upload all main figures as separate Figure files in .tif or .eps format. For more information about how to convert and format your figure files please see our guidelines:

6) We notice that your supplementary Figures, and Tables are included in the manuscript file. Please remove them and upload them with the file type 'Supporting Information'. Please ensure that each Supporting Information file has a legend listed in the manuscript after the references list.

7) In the online submission form, you indicated that The data utilized in this study are available upon reasonable request.. All PLOS journals now require all data underlying the findings described in their manuscript to be freely available to other researchers, either

- In a public repository

- Within the manuscript itself

- Uploaded as supplementary information.

8) Please ensure that the funders and grant numbers match between the Financial Disclosure field and the Funding Information tab in your submission form. Note that the funders must be provided in the same order in both places as well.

Reviewers' Comments:

Reviewer's Responses to Questions

Key Review Criteria Required for Acceptance?

Methods:

-Are the objectives of the study clearly articulated with a clear testable hypothesis stated?

-Is the study design appropriate to address the stated objectives?

-Is the population clearly described and appropriate for the hypothesis being tested?

-Is the sample size sufficient to ensure adequate power to address the hypothesis being tested?

-Were correct statistical analysis used to support conclusions?

-Are there concerns about ethical or regulatory requirements being met?

Reviewer #1: Goals of the study are clearly stated, though they seem to overstate what the study was able to accomplish in that comprehensive characterization of behavior is challenging from only two time points and in that the quantification of exposure to fecal contamination is difficult from indicator bacteria given the limitations of said indicator.

Ethiopian specific vocabulary needs to be defined (i.e., kebeles, injera, etc.) in order to cater to a larger audience.

Sample size is strong, though the authors need to address the lack of soil data from time point 1.

The decision to consider drinking a duration behavior compared to eating as a frequency-based behavior needs justification. Samples are listed as clear liquids that should not be considered clear liquids.

While a previous publication is referenced, the author still needs to provide a more thorough overview of methodology including the limitations cited in the external paper (i.e., sample volume, lack of sterilization of caregiver hands prior to breast milk collection, pathogenic vs. nonpathogenic E. coli, etc.). In the other study, it references positive controls but not negative controls or the results of said controls. This needs to be included in the work.

Would appreciate a discussion of whether or not seasonality played a role in environmental contamination risks.

The authors should provide justification for the use of a normal distribution for concentration data. Did they carry out statistical analysis on their human/environmental sampling data to determine this?

The authors need to include the calculations for exposure, including whether or not transfer efficiencies were used, how surface areas of touching and mouthing behaviors were considered, how were timeseries behaviors accounted for, etc.

Reviewer #2: 1. The methods lack the clarity required to repeat the study. While the details may be in the github code, it should not be the responsibility of the reader to comb through the code to determine critical information such as the transfer efficiencies used in the modules to determine the concentration of E. coli on the hand before and after specific transitions.

2. The methods also exclude critical details of the modules such as soil not being assessed at Timepoint 1 and direct contact with feces not included at Timepoint 1 or Timepoint 2. The authors make strong conclusions that food and breastfeeding as primary pathways in Timepoint 1 without acknowledging that soil could be a primary pathway but it was not considered at Timepoint 1.

3. The methods section on the modules should also explain how the frequency of events was translated into the quantity of food/soil ingested. Given that the E. coli was measured as concentration per volume, it is necessary to estimate the volume of food/soil ingested to estimate the quantity of E. coli ingested. The quantity of E. coli ingested cannot be estimated from the frequency alone. Without a valid methodology for this conversion, the method cannot be considered valid.

4. Figure 2: There is an arrow from NI(i) to NI(i+1) but it is not clear why NI(i) impacts NI(i+1). This aspect of the model should be explained in the methods.

5. Given the potential importance of direct fecal ingestion, the authors should justify why this activity was not included in the list of observed behaviors.

6. It appears that the behavioral categories and compartments used are not comprehensive and mutually exclusive. This both confuses the presentation and raises questions about what is being modeled. For example, Table 1 presents the percentage of time awake, bathing, drinking, and sleeping. Children should be awake while bathing and drinking, so the percentage of time awake should be inclusive of the time bathing and drinking. One way to resolve this is to describe the time “awake” as the time “playing” or “awake but not bathing and drinking”. Table 1 also lists the compartments of “carried by mother”, “carried by other”, “on the bare ground”, and “on a surface with barriers”. How is the time classified when the child is awake but on a bed or chair? Awake but on a mat? What is the definition of a “surface with barriers”? Furthermore, why was “touching soil” not included? If the model seeks to represent the E. coli contamination on hands, then the frequency of touching soil indoors on a mat, indoors on the ground, and outdoors should be included in the model, along with the respective E. coli on these surfaces (instead of just one concentration for E. coli on “soil”). Having lists of activities and compartments that are comprehensive and mutually exclusive is essential given the desire to use a competing hazard model.

7. The authors assumed that children were awake for 14 hours at both Timepoint 1 and Timepoint 2. Instead, they should use empirical data (from their study or others) to estimate the actual time awake. Other studies have found that as children age, they sleep less.

8. What does “pica – other” include?

9. Figure 4: The authors should describe in the methods why they consider food to be a vector instead of a source and how this affects the model. Since we know that contamination in food can grow over time, the authors should also consider a sensitivity analysis that treats the food as a source instead of a vector. Similarly but likely less importantly, bathing water should be considered as a source in some cases – e.g. a drop of bathing water placed in the mouth of an infant is acting as a source, whereas the child’s skin going into the bathing water has the water acting as a sink.

10. Lines 99-101: Explain how many local enumerators there and what standard of inter-rater reliability they had to achieve before the started collecting data. During data collection how often was inter-rater reliability re-assessed and what did you do with data that did not meet the standard?

Reviewer #3: Yes, I suggest additional clarity on some statistical test results described, and also some more statistical tests suggested.

Results:

-Does the analysis presented match the analysis plan?

-Are the results clearly and completely presented?

-Are the figures (Tables, Images) of sufficient quality for clarity?

Reviewer #1: In lines 197-199, the sentence structure needs to be re-written to be clear what the percentage is specifying. As currently written, it very unclear what the percentage refers to.

When discussing correlations, the author should address strength of correlation via numerical values.

The author needs to not refer to positive E. coli samples as fecal contamination. Not all E. coli is fecal contamination and much of the E. coli found in the environment is non-pathogenic. The author should use specific language to specify either E. coli contamination or an indication of possible fecal contamination.

It feels that the results section could be more profound. Some of the results section was previously reported, making the portion that is newly explored minimal. Are there are other ways the author could look at the data to provide a more robust set of results? For example, looking at cumulative exposure over a given day (or other time period) in addition to just exposure pathways.

The last sentence of the results section seems to be contradicted by the data and the discussions section with the environment (soil) - hand - mouth pathway being substantially important at time point 2.

Reviewer #2: 1. Table 1 should include the percentage of children who could crawl, since crawling strongly influences mouthing and touching behaviors.

2. Figure 3 contains several boxes that are not labeled – what are these?

3. Table S1 is misleading and not clear. Firstly, what does “collection volume” mean? The areola sample was collected with a swab. Does “collected” volume mean “elution volume? Secondly, the table states that up to 1 L of sample was processed, but Deblais 2024 reports that a maximum of 15.84 mL of sample was processed. This results in a much lower limit of detection than would be expected by the reader given the 1 L of sample collected. It is unclear why 1 L of sample was collected when only 15 mL was assays (especially given the limited availability of drinking water at households), or, conversely, why so little sample was assayed without prior concentration of the sample. To clarify, Table S1 should include the lower limit of detection in MPN and CFU. Table 1 should also note the approximate surface area represented by the sample (e.g. for bootsocks, what is the approximate m2 that was covered with “15 steps” (Deblais 2024). The table (and methods section) should also specify is the “soil” referred to in the paper is from inside or outside of the households, as Deblais 2024 notes that three soil samples were collected, two from inside and one from outside. Presenting the surface areas represented will help determine the load, allowing for comparison to other studies.

4. Figure S4 – the results state that children commonly took naps at noon at Timepoint 2 but this is not apparent from Figure S4.

Reviewer #3: Yes, I provide some details where additional information is suggested.

Conclusions:

-Are the conclusions supported by the data presented?

-Are the limitations of analysis clearly described?

-Do the authors discuss how these data can be helpful to advance our understanding of the topic under study?

-Is public health relevance addressed?

Reviewer #1: The discussion (and conclusions) is supported by the data present (with the exception of the last sentence of results addressed above in the Results section).

The studies for comparison to support frequent geophagia (soil pica) were not strong support.

The authors should elaborate on the impacts of the missing observation time. Are there thoughts on this time being higher or lower impact for exposure? All behavior is somewhat limited by observation time because behaviors cannot be observed at all times, so this limitation should be caveated as minor, observations were long.

In the Conclusions, the month range is specified more clearly. This should be in methods.

The authors need to address other limitations of the study (which were alluded to above) - especially the idea of indicator organisms not necessarily being pathogenic and the role of environmental reservoirs of nonpathogenic E. coli. Another limitation that might be worth discussing is the idea of multi-pathogen exposure that cannot be captured by only one indicator organism.

Support the public health relevance for interventions and the future research into more specific enteric pathogens.

Reviewer #2: The conclusions are not specific to the data, but rather seem speculative / based on the results of other papers.

Some specific comments on the intro and discussion section:

1. The discussion could be stronger by comparing the frequency of activities with those found in prior time-activity studies in LMICs and high-income countries. To facilitate this comparison, the rate of any drinking, any eating, any mouthing, and any pica should be listed in Table 2 and presented in Figure S1.

2. The discussion could also be improved by more specifically discussing data that is found in the results and not discussing data that is not found in the results. For example, the discussion makes some mention of the need for animal feces management, but presents no data that the soil is contaminated by poorly managed animal feces and does not refer to the literature from some studies that have found outdoor soil contamination to be associated with the ownership of free-roaming animals (e.g. Ercumen 2017 – animal feces contribute to domestic contamination)

3. The concentration of touching fomites seems low compared to prior studies. The discussion should note why the authors think this was found.

Reviewer #3: (No Response)

Editorial and Data Presentation Modifications?

Reviewer #1: See above comments. In addition to the above comments, the author should reconsider the title because it is misleading to specify fecal contamination when an indictor organism is used. Would recommend specifying E. coli or something along those lines. Further, in the introduction, the author talks about the study focusing on Campylobacter seems unnecessary as they never return to this pathogen and the study does not actually look for Campylobacter but an indicator organism, which has unproven status as an indicator in this setting.

Reviewer #2: Line 45: Instead of saying “basic WASH”, describe what these interventions included so it is more clear what did not work.

Reviewer #3: See Summary Comments

Summary and General Comments:

Reviewer #1: The paper is an interesting exploration of pathway routes, and the rural landscape is under-explored, making it a worthwhile contribution to the existing body of literature on the topic. As such, the paper should be published but needs significant revision to expand on the methods, the limitations of the study, and to flesh out additional results to improve the robust contributions it makes to literature.

Reviewer #2: 1. Some references are inapproriately old (e.g. references from 2012 and 2016 are used in the introduction to introduce the burden of diarrhea).

2. Relevant references from other time-activity studies in LMICs are missing and should be included in the intro and discussion section (e.g. Ngure 2013 – geophagy and ingestion of animal feces, Mattioli 2015 – hand-to-mouth contacts vs drinking water, Kwong 2020 - ingestion of fecal bacterial along multiple pathways)

Reviewer #3: The study provides a novel dataset based on observations of child-environment interactions for children in vulnerable age windows of 4-8 months and 11-15 months in rural Ethiopia.

The data are integrated with previously published environmental contamination data using a modeling approach to estimate children's exposures to E. coli through the environment, and highlight child-environment interactions that lead to the greatest risk of disease.

The study is thoughtful and thorough, and well written. The findings are useful for helping to guide decisions around hygiene interventions, and support the need for better access to water, sanitation, and hygiene infrastructure.

Major Comments:

1) The study replicates some earlier work on child-environment observations in other contexts, including the described potential use case of monitoring chemical exposures. The introduction would benefit from an overview of previous work, to highlight that this study -while innovative- is rooted in previous studies using structural observation to understand environmental exposures. How do child behaviours observed here compare to other studies, or are they not comparable due to not standardized methods?

2) The modeling work (Section 2.4) is only discussed in minimal detail, with a reference to the availability of the code. Although this approach is maybe o.k., I think an important topic to discuss is what influence modeling assumptions will have on the major conclusions from the model. For example, are there modeling assumptions that reduce the relative contributions of hand-to-mouth contacts for children? I am very surprised by this finding, given others (see Mattioli et al. ) have shown hand-to-mouth contacts are quite important, and I'm curious what drives this results, modeling framework or observations. More generally: what model assumptions influence outcomes and their interpretation?

Minor Comments:

1) Abstract should be readily understood, so I suggest describing Countee as no one will be familiar with this, and also replacing Timepoint 1 and Timepoint 2 with the age ranges of the children (4-8 mo; 11-15 mo, if I understand correctly). Generally replace Timepoint 1 and 2 with age ranges throughout whole manuscript to improve readability.

2) 32-34 - these are old numbers, are there updated numbers available?

3) 46-51 - consider reframing, the argument here is not clear. For example, you could establish that a mechanistic explanation of the conclusions of the RCTs would improve understanding; and fecal contamination along exposure ways may help to establish or validate a mechanistic explanation for why interventions do/do not reduce exposures and subsequent diarrhea.

4) 93 - describe CAGED trial if referring to it is important; audience will not be familiar with it.

5) 102-109: a comment would be helpful to describe the degree to which these specific categories of activities are location specific, or sufficiently generalizable.

6) 166 - child's hands?

7) Can you use and describe statistical tests to compare results between the two time points (Table 1 results), as differences are discussd in lines 196-201.

8) 228 - give stats test and metrics when stating something is statistically different.

9) 249 - what assumptions in the modeling framework influence this? Why does this deviate from Mattioli et al. findings? DOI: https://doi.org/10.1021/es505555f. See major comment.

10) 262 - how is this well calibrated?

11) 274-298 - some assertions without references made, suggest supporting statements in this section with evidence.

12) 284-288 - it is well established that diarrheal disease risks increase for older children compared to younger children; suggest connecting your findings to those studies.

13) 311-315 -authors neglect to describe their results in the context of other studies on structured observations of children. Suggesting linking results to other published results. See major comments.

14) 344 - multiple studies looking at children's exposures to environments have been done to characterize chemical exposures (see for example U.S. EPA Children's Exposure Factors Handbook). Suggest authors better describe this in the introduction (see Major Comment)

15) 365 - new paragraph?

16) 360 - Timepoints are clearly linked to age ranges here; suggest using age ranges, not time points (see minor comment 1).

17) Discussion - comment on the degree to which these findings for children are specific to this location, or are generalizable to other settings and scenarios. Are we supposed to conduct similar studies everywhere we want to decide WASH interventions, or can these findings be applied to new areas?

PLOS authors have the option to publish the peer review history of their article (what does this mean?). If published, this will include your full peer review and any attached files.

Do you want your identity to be public for this peer review? For information about this choice, including consent withdrawal, please see our Privacy Policy.

Reviewer #1: No

Reviewer #2: No

Reviewer #3: No

Figure resubmission:
---

## [Editor Report · Decision Letter 1]

Dear Dr. Wang,

We are pleased to inform you that your manuscript 'Quantitative Multi-pathway Assessment of Exposure to Escherichia coli for Infants in Rural Ethiopia' has been provisionally accepted for publication in PLOS Neglected Tropical Diseases.

Best regards,

Josh M Colston, Ph.D.

Academic Editor

Ana LTO Nascimento

Section Editor

Shaden Kamhawi

co-Editor-in-Chief

Paul Brindley

co-Editor-in-Chief

Thank you for your thorough and considered responses to the reviewers' comments. The already strong manuscript has been much improved as a result and is worthy of publication in PLOS NTDs. Congratulations.

---

## [Editor Report · Acceptance letter]

Dear Dr. Wang,

We are delighted to inform you that your manuscript, "Quantitative Multi-pathway Assessment of Exposure to Escherichia coli for Infants in Rural Ethiopia," has been formally accepted for publication in PLOS Neglected Tropical Diseases.

Best regards,

Shaden Kamhawi

co-Editor-in-Chief

Paul Brindley

co-Editor-in-Chief
